# Top predator reveals the stability of prey community in the western subarctic Pacific

Dongming Lin[1,2,3,4], Xinjun Chen[1,2,3,4,5]*

**1** College of Marine Sciences, Shanghai Ocean University, Shanghai, China, **2** Key Laboratory of Sustainable Exploitation of Oceanic Fishery Resources, Ministry of Education, Shanghai, China, **3** National Distant-water Fisheries Engineering Research Center, Shanghai, China, **4** Key Laboratory of Oceanic Fisheries Exploration, Ministry of Agriculture and Rural Affairs, Shanghai, China, **5** Laboratory for Marine Fisheries Science and Food Production Processes, Qingdao National Laboratory for Marine Science and Technology, Qingdao, China

* xjchen@shou.edu.cn

**Data Availability Statement:** All relevant data are within the paper and its Supporting Information files.

## Abstract

The stability of the ecosystems depends on the dynamics of the prey community, but changes in the composition and abundance of prey species are poorly understood, especially in open ocean ecosystems. We used neon flying squid *Ommastrephes bartramii*, an active top predator, as a biological sampler to investigate the dynamics of the prey community in the southwestern part of the Western Subarctic Gyre in the northwestern Pacific Ocean. Squid were collected monthly from July to November 2016. There were no significant differences among months in stable isotopes ($\delta^{13}$C and $\delta^{15}$N) in the digestive gland, a fast turnover organ reflecting recent dietary information. Similar findings were obtained from analyses of isotopic niche width and fatty acid profiles. The potential influence of the environment (monthly mean sea surface temperature, SST, and chlorophyll-*a*, Chl-*a*) on the prey community was examined with SST and Chl-*a* both varying significantly among sampling months. We found little evidence for significant effects of SST and Chl-*a* on the isotopic values, nor on the fatty acid profiles except for 20:4n6 and 24:1n9. These lines of evidence indicate that the prey community in the southwestern part of the gyre remains stable, with little evidence for systematic changes at the community level. This study provides a novel understanding of the dynamics of the prey community and highlights the use of top predators to study the trophic dynamics of an oceanic system where a long-term scientific survey is unavailable.

## Introduction

Concern about ecosystem functioning [1–3] highlights the need for a better understanding of how the composition and abundance of species in natural communities respond to environmental change. For instance, predatory animals are susceptible to reduction or extirpation of available prey due to environmental processes, which undermines the stability of ecosystems and the services they provide [3, 4]. Stability is central to ecosystem functioning, which includes the ecological processes controlling the fluxes of energy, nutrients and organic matter

**Funding:** This work was supported by National Key Research and Development Project (2019YFD0901404), National Natural Science Foundation of China (41876144, 41876141), Natural Science Foundation of Shanghai (16ZR1415400) and Shanghai Science and Technology Innovation Program (19DZ1207502). The funders had no role in study design, data collection and analysis, decision to publish, or preparation of the manuscript.

**Competing interests:** The authors have declared that no competing interests exist.

[5–7]. Therefore, insight into the prey community and its stability is important for a comprehensive understanding of how an ecosystem responds to ongoing environmental change [8]. In the northwest Pacific Ocean, large areas are highly productive and support large populations of pelagic predators including squids [9–11]. Previous studies in this region have identified that oceanographic productivity is significantly driven by spatial-temporal variation of the anticyclonic and cyclonic gyres [12], which greatly influences the abundance of higher trophic level predators [9, 13]. However, the status of the prey community that supports higher trophic level predators is poorly studied, which limits our understanding of the functioning of the overall northwest Pacific ecosystem. The Western Subarctic Gyre, a cyclonic gyre in the northwest Pacific Ocean, is one region with limited scientific monitoring and hence understanding of the dynamics of prey communities, even though this information is needed for assessing ecosystem functioning.

Squids grow rapidly, have short lifespans, and semelparous reproduction [14, 15]. They impose considerable predation pressure on low- and mid-trophic level species [16] due to their voracious and active feeding [17–19], and simultaneously support the productivity of other predators [20]. They consequently play a key role in ecosystem functioning [20, 21]. Squid are highly adapted to the environment to exploit a diverse range of prey and habitat resources [22, 23]. They occupy medium to top trophic positions in many marine food webs and their trophic niche width differs among species and ecosystems [24, 25]. These characteristics reflect not only their flexible feeding strategy [23, 26], but also provide information on the trophic structure of the system in which they are found [27, 28]. Increasingly, squid have been highlighted as indicators to examine major changes in trophic structure and ecosystem functioning [20, 26, 29].

Many naturally occurring biochemical tracers such as stable isotopes and fatty acids have increased the ability to quantify and characterize complex food webs and community dynamics [24, 30, 31]. These techniques can assess a predator's dietary history over a range of temporal scales, reflecting "you are what you eat" [32, 33]. Biochemical tracers are considered to be a complementary or even alternative and cost-effective tool to stomach content analysis for examining major changes in trophic structure and ecosystem productivity [26, 34]. For example, Pethybridge et al. [26] reported that the comparison of fatty acid profiles of *Todarodes filippovae* with those of its potential prey taxa revealed temporal dietary shifts related to site-specific oceanography and ecosystem structure in continental slope waters in the Southern Ocean. Stable isotope ratios of nitrogen ($\delta^{15}N$) and carbon ($\delta^{13}C$) for higher trophic organisms match of those of their prey [32, 34]: $\delta^{15}N$ values are enriched by about 3‰ per trophic level, while $\delta^{13}C$ values change little among trophic levels in marine food webs. It is possible to estimate the trophic width of species, populations and ecosystems by analyzing $\delta^{15}N$ and $\delta^{13}C$ data together [34–36]. In relation to fatty acids, marine heterotrophs are subject to biochemical limitations in biosynthesis and modification of carboxylic acids, and generally assimilate dietary fatty acids with little or no modification [37]. Many individual fatty acid tracers (e.g., 20:4n6, 20:5n3, 22:6n3) have been used to study trophic ecology and have revealed the overlapping influences of temperature, habitat, trophic guild and phylogeny (see Meyer et al. [38]). Thus, by selecting an appropriate predator, stable isotopes and fatty acids could allow the estimation of trophic structure and its dynamics at multiple time scales.

We use neon flying squid *Ommastrephes bartramii* as a biological sampler to investigate the trophic dynamics of the prey community of the southwestern part of the Western Subarctic Gyre in the northwest Pacific Ocean. This region is characterized by high productivity that supports a large population of higher trophic level species including *O. bartramii* [9, 39]. *O. bartramii* is an extremely widely distributed ommastrephid with a worldwide oceanic bi-subtropical distribution, and inhabits the entire water column through the epipelagic, mesopelagic

and upper bathypelagic zones [15]. More importantly, *O. bartramii* is a high trophic level species, with an average $\delta^{15}$N value up to 13.6‰ [17, 19, 40, 41], which occupies a similar trophic position as other top predators such as albatrosses (mean $\delta^{15}$N, 12.0‰ for *Diomedea immutabilis*; 14.4‰ for *Diomedea nigripes*) [40], and sharks (*Prionace glauca*, mean $\delta^{15}$N 12.1‰) [42]. *O. bartramii* is an opportunistic generalist that preys on a wide variety of species, including crustaceans, fishes and cephalopods [15, 17, 18, 43]. The diet of *O. bartramii* varies spatial-temporally given the associated prey community, e.g., it may feed on transitional-water species during its northward feeding migration [18], migratory mesopelagic species in the epipelagic zone at night [44], and non-migratory species during the day in the mesopelagic zone [18]. *O. bartramii* therefore has the potential to be an ideal trophic indicator of ecosystem functioning [14], and represents a way of integrating ecological dynamics over a large area and across several ecosystems that are difficult to study directly [4]. We analyzed carbon ($\delta^{13}$C) and nitrogen ($\delta^{15}$N) stable isotope ratios and fatty acids from the digestive gland of *O. bartramii*—the digestive gland having been shown to provide information on recent diet (10–14 days) of cephalopods [45–48].

We aim to (a) determine the isotopic trophic niche and variation of the prey community of *O. bartramii*; and (b) assess the dynamics of the prey community over a relatively long period. These results will increase our understanding of the systematic changes in the ecological community in the region, and provide a basis for quantifying community dynamics in response to environmental change.

## Materials and methods

### Ethics statement

Specimens were collected as dead squids from the small-scale trawl fishery landings, from July to November 2016. The specimens were analyzed in the laboratory using methods consistent with current Chinese national standards, namely Laboratory Animals—General Requirements for Animal Experiment (GB/T 35823–2018). There was no requirement for ethics approval of sampling protocols because all the material analyzed in this paper were obtained from commercial fishermen and were already dead.

### Study area

The Western Subarctic Gyre is the western cyclonic subgyre in the North Pacific Ocean and is found in the northern Kuroshio-Oyashio transition zone [49]. It is nutrient rich owing to upwelling, presumably due to the Oyashio Current in the southwest and Subarctic Current in the south [49, 50]. It has shallow mixed depth and photic zone [51, 52]. The phytoplankton biomass is maximal during spring and does not differ significantly during summer, autumn and winter [53]. The zooplankton community is relatively simple [54], and the biomass assemblage is dominated by large interzonal copepods [55]. It is supposed that microzooplankton and other mesozooplankton taxa replace phytoplankton as the primary food source for dominant mesozooplankton species, which are then preyed on by micronekton and larger zooplankton [54].

### Biological data collection

*Ommastrephes bartramii* were collected monthly from July to November 2016 from commercial fishing operations in the Western Subarctic Gyre (see the sample stations in Fig 1). This period is considered to be one of active feeding and growth for the winter-spring cohort in the northwest Pacific Ocean [15]. The specimens were frozen immediately onboard under -30°C,

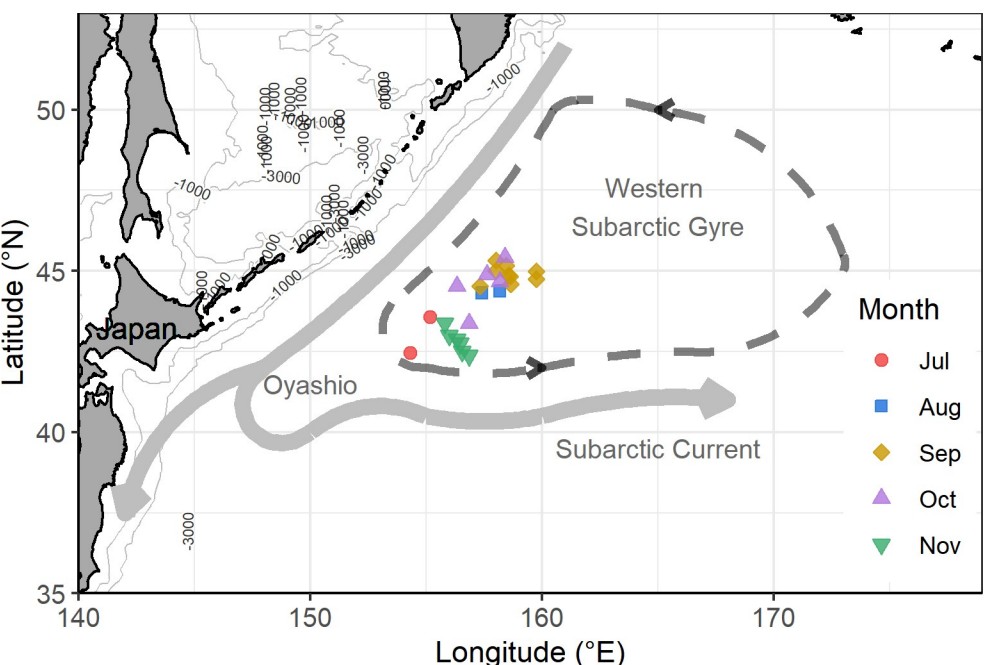

**Fig 1. Study area showing sample locations and selected bathymetric contour lines in the western subarctic Pacific with a schematic illustration of western subarctic gyre and the near-surface current.** The schematic diagram of Western Subarctic Gyre and its currents is redrawn from Qiu [49].

and shipped to the laboratory for further analyses. After defrosting at room temperature, 129 specimens that covered all the sampling months were randomly selected (Table 1). Dorsal mantle length (ML, 1 mm), body weight (BW, 1 g), and sexual maturation were recorded for each specimen. Macro-scale maturity stages were assigned following [56], and all specimens were found to be subadults with developing and maturing gonads.

The whole digestive gland of each selected specimen was dehydrated in a freeze-drying chamber (Crist Alpha 1-4/LDplus; (Martin Christ Gefriertrocknungsanlagen GmbH, Germany). After dehydration, each digestive gland was ground up in a mortar and pestle, and a subsample (~1 g) was taken for fatty acid and stable isotope analysis.

## Fatty acid analysis

The subsample of each digestive gland was extracted using a 2:1 (v/v) chloroform:methanol solution [57]. The lipids were used for fatty acid analysis, while the lipid-extracted samples were lyophilized again for at least 24 hours for stable isotope analysis.

**Table 1. Mantle length and body weight of *O. bartramii* from which samples of digestive gland were taken.**

| Sampled month | N | Mantle length (ML, mm) | | Body weight (BW, g) | |
|---|---|---|---|---|---|
| | | Range | mean±SD | Range | mean±SD |
| July | 33 | 194; 283 | 225.39±20.82 | 163; 795 | 333.73±136.58 |
| August | 16 | 209; 275 | 251.56±20.3 | 246; 628 | 436.19±122.91 |
| September | 20 | 236; 288 | 261.4±13.39 | 408; 780 | 513.75±83.49 |
| October | 26 | 219; 345 | 278.69±44.47 | 312; 1274 | 667.23±305.07 |
| November | 34 | 218; 370 | 277.41±33.19 | 302; 1491 | 622.88±232.46 |
| pooled | 129 | 194; 370 | 258.67±36.35 | 163; 1491 | 517.78±237.94 |

The extracted lipids were used for fatty acid determination following the "Determination of total fat, saturated fat, and unsaturated fat in foods—Hydrolytic extraction-gas chromatography" [58] protocol. Fatty acid methyl esters (FAME) were analyzed separately for each sample using an Agilent 7890B Gas Chromatograph (GC) coupled to a 5977A series Mass Spectrometer Detector (MSD, Agilent Technologies, Inc. USA). The fatty acid 19:0 was used as an internal standard. The separation was carried out with helium as the carrier gas, and a thermal gradient programmed from 125°C to 250°C, with an auxiliary heater at 280°C. The total fatty acids were determined as dry tissue weight (mg/g dry weight), and each fatty acid was expressed as a percentage of total fatty acids in the sample [37].

## Stable isotope analysis

Due to contaminants when lyophilized again, 56 lipid-extracted subsamples of the digestive gland were not used for stable isotope analysis. Consequently, a total of 73 lipid-extracted subsamples were used and ground separately to a homogeneous fine powder, and a ~1.0 mg subsample for each subsample was used for stable isotope analysis. Stable isotope ratios ($\delta^{13}$C and $\delta^{15}$N) were measured separately for each sample using an IsoPrime 100 isotope ratio mass spectrometer (IsoPrime) and vario ISOTOPE cube elemental analyzer (Elementar Analysensysteme). The standards for carbon and nitrogen followed Gong et al. [59]: using international reference materials (USGS 24 [$\delta^{13}$C = −16.049‰], USGS 26 [$\delta^{15}$N = 53.7‰]) and the laboratory running standard (protein [$\delta^{13}$C = −26.98‰ and $\delta^{15}$N = 5.96‰]). The measurement errors were approximately 0.05‰ and 0.06‰ for $\delta^{13}$C and $\delta^{15}$N, respectively.

## Statistical analysis

Isotopic values and fatty acids were tested for significant differences between sampling months. All data were first checked for normality using the one-sample Kolmogorov-Smirnoff test and for homogeneity of variances using Levene's test [60]. One-way ANOVA was then applied to test for differences, and a Tukey's post hoc test [60] performed to determine where the difference occurred when significant differences were found. Data were analyzed using a Kruskall-Wallis nonparametric one-way ANOVA test and a Games-Howell post hoc test [60] when normality and/or homoscedasticity were rejected.

Stable Isotope Bayesian Ellipses (SIBER) [61] implemented in R [62] were used to analyze the stable isotope data in the context of isotopic niche between sampling months. We calculated the prey community isotopic niche widths for each sampling month, including the standard ellipse area (SEAb), the corrected standard ellipse area (SEAc, an ellipse containing 40% of the data regardless of sample size) and the overlap as the proportion of the sum of the non-overlapping ellipse areas (non-overlap SEAc proportion) based on 1,000 replications [61]. The non-overlap SEAc proportion ranges from 0 (completely distinct ellipses, indicating zero overlap in the isotopic niche widths between groups), to 1 (completely coincidental ellipses, indicating a complete overlap in the isotopic niche widths between groups) [61]. SEAb was used to test for differences in the isotopic niche area of the prey community between months, while SEAc and the non-overlap SEAc proportion were used to compare the niche width of the prey community over months. These analyses allowed the trophic dynamics of the prey community to be identified.

Non-metric multidimensional scaling (nMDS) and analysis of similarities (ANOSIM) were applied to assess the similarities of fatty acid profiles between months. These analyses could allow for the identification of potential differences in the trophic structure of the prey community among months, similar to the analyses of dietary data for a specific species [48]. Each fatty acid was expressed as a percentage of total fatty acids, and a square-root transformation was used to avoid over-emphasis of extreme values [37]. A Bray–Curtis dissimilarity measure was

employed in the nMDS and ANOSIM [63, 64]. The analyses were performed in the package 'vegan' [65] in R.

Generalized additive mixed models (GAMMs) [66] with sampling month as the random effect were used to access the potential effects of the ambient environment on the dynamics of the prey community. This involved testing for potential relationships between isotopic values, fatty acids, and the environmental variables. The dependent variables were $\delta^{13}C$, $\delta^{15}N$, and the individual fatty acid that was found to differ significantly between sampling months. Key predictors were monthly mean sea surface temperature (SST, ˚C) and chlorophyll-*a* concentration (Chl-*a*, mg m-3). SST and Chl-*a* were downloaded from the National Oceanic and Atmospheric Administration (NOAA) ERDDAP (Version 1.82) (https://oceanwatch.pifsc.noaa.gov/erddap/index.html), at a resolution of 0.05˚× 0.05˚. Preliminary analysis indicated that both SST and Chl-*a* were not correlated with each other (variance inflation factor = 1.54). The effect of sampling month was taken to be random to account for temporal effects in the data and unexplained differences among the prey community. We used the function 'gamm' with a Gaussian error distribution in the package 'gamm4' [67] in R.

## Results

The sampled *O. bartramii* ranged from 194 to 370 mm ML and from 163 g to 1491 g. Body size increased significantly with sampling month (ML, $F = 16.99$, $P<0.05$; BW, $F = 13.35$, $P<0.05$) (Table 1).

### Stable isotopic and niche analyses

$\delta^{13}C$ ranged between -22.18‰ and -19.13‰, with an average of -20.49 ± 0.70‰, and $\delta^{15}N$ ranged between 5.18‰ and 9.88‰, with an average of 8.42 ± 0.96‰ (Table 2). The highest values of $\delta^{13}C$ and $\delta^{15}N$ occurred during October, but no significant differences in $\delta^{13}C$ and $\delta^{15}N$ were detected among months (Kruskal-Wallis test, $\delta^{13}C$, $\chi^2 = 0.84$, $P = 0.93$; $\delta^{15}N$, $\chi^2 = 4.99$, $P = 0.29$). The variation of $\delta^{13}C$ between the minimum and the maximum values was similar among months (range 2.60‰ to 3.02‰). Similar findings were obtained for $\delta^{15}N$, where the variation ranged from 3.27‰ to 4.49‰ (Table 2).

The Bayesian isotopic niche analyses did not find significant differences in the standard ellipse area (SEAb) among months (Kruskal-Wallis, $\chi^2 = 6.26$, $P = 0.18$) (Fig 2A). The corrected standard ellipse area (SEAc) ranged from 1.36 to 1.50, and indicated considerable

**Table 2. Stable isotopic values and isotopic niche width metrics.** Isotopic values were determined from the digestive gland of *O. bartramii*. The variation between the minimum and the maximum isotopic values is given in parenthesis under the ranges. SEAc, corrected standard ellipse area; non-overlap SEAc proportion, proportion of the sum of the non-overlapping ellipse areas.

| Sampling month | N | δ13C (‰) | | δ15N (‰) | | Isotopic niche width | |
|---|---|---|---|---|---|---|---|
| | | Range | mean ± SD | range | mean ± SD | SEAc | non-overlap SEAc proportion |
| July | 12 | -22.04; -19.41 (2.63) | -20.57 ± 0.77 | 6.59; 9.88 (3.29) | 8.46 ± 0.88 | 1.46 | |
| | | | | | | | 0.70 |
| August | 15 | -22.09; -19.49 (2.60) | -20.54 ± 0.67 | 5.18; 9.67 (4.49) | 8.38 ± 1.04 | 1.36 | |
| | | | | | | | 0.76 |
| September | 16 | -22.15; -19.37 (2.78) | -20.46 ± 0.74 | 6.12; 9.37 (3.27) | 8.47 ± 0.91 | 1.43 | |
| | | | | | | | 0.77 |
| October | 18 | -22.15; -19.13 (3.02) | -20.37 ± 0.68 | 5.72; 9.79 (4.07) | 8.65 ± 1.05 | 1.41 | |
| | | | | | | | 0.72 |
| November | 12 | -22.18; -19.57 (2.61) | -20.53 ± 0.71 | 5.60; 9.37 (3.77) | 8.15 ± 0.99 | 1.50 | |
| Pooled | 73 | -22.18; -19.13 (3.05) | -20.49 ± 0.70 | 5.18; 9.88 (4.70) | 8.42 ± 0.96 | - | - |

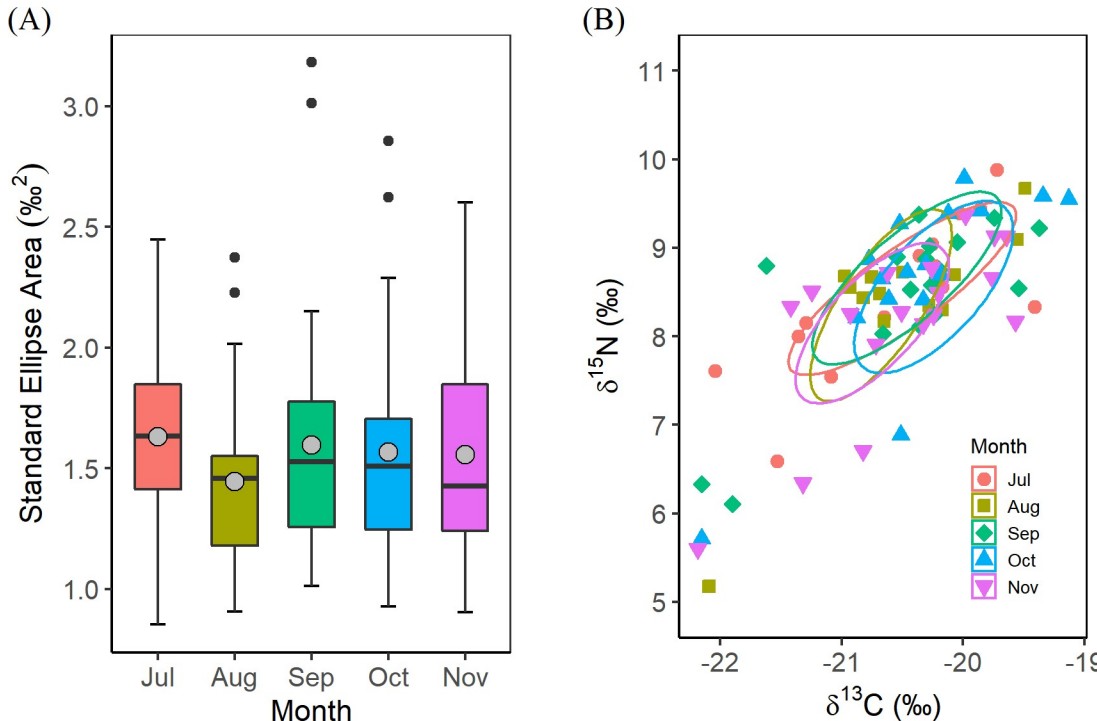

**Fig 2. Estimates of isotopic niche area for each sampling month based on $\delta^{13}C$ and $\delta^{15}N$.** Isotopic values were determined from the digestive gland of *O. bartramii*. (A) Bayesian standard ellipse area (SEAb) estimates for each sampling month. The boxes cover the central 50% of the distributions and bars the 90% intervals, with grey solid circles and horizontal lines indicating the means and medians; black points indicate whiskers. (B) 40% Bayesian credible intervals for the standard ellipse for each sampling month.

overlap in the isotopic data between each consecutive sampling month, confirmed by the high non-overlap SEAc proportions (Table 2) and the considerable overlap of the ellipses for the different months (Fig 2B).

## Fatty acids and dissimilarity analyses

No significant difference in the total fatty acids was found among months (ANOVA, $F = 1.56$, $P = 0.19$) (Table 3). Similarly, no significant differences in the proportions of the main fatty acid classes were detected among months, except for the saturated fatty acids (SFA) where July was significantly higher than the remaining months (Tukey HSD, $P < 0.05$) (Table 3). In terms of individual fatty acid profiles, 16 fatty acids varied significantly between months (11:0, 13:0, 17:0, 18:0, and 16:1n7 higher in July; 20:1, 22:1n9 and 20:5n3 higher in September, 24:1n9 and 20:2 higher in October, and 14:1n5, 18:2n6t, 18:3n6, 20:3n6, 20:4n6 and 22:2n6 higher in November; Table 3).

In contrast, nMDS revealed a considerable overlap in the overall fatty acid profiles (Fig 3). These findings were confirmed by ANOSIM, in which the dissimilarity value (ANOSIM statistical *R* value) between each two consecutive months ranged from 0.08 to 0.18, with a global value of 0.15 for all months pooled (Table 4).

## Potential relations to sea surface environments

Monthly mean sea surface temperature (SST) at the sampling stations varied significantly among months ($F = 38.19$, $P < 0.01$), with the highest temperature in August (mean±SD, 19.83

**Table 3. Fatty acids in the digestive gland of *O. bartramii* sampled in the western subarctic gyre of northwest pacific ocean, from July to November 2016.** SFA, saturated fatty acids; MUFA, monounsaturated fatty acids; PUFA, polyunsaturated fatty acids; TFA, total fatty acids. Values are mean ±SD; different superscript letters within rows represent significant differences ($P<0.05$) detected using the post hoc test.

| Terms | July | August | September | October | November |
|---|---|---|---|---|---|
| *Fatty acid (%ΣTFA)* | | | | | |
| 10:0 | 0.06±0.01 | 0.06±0.04 | 0.05±0.01 | 0.05±0.02 | 0.05±0.04 |
| 11:0 | 0.25±0.09[b] | 0.16±0.05[a] | 0.13±0.08[a] | 0.19±0.08[ab] | 0.18±0.06[a] |
| 12:0 | 0.14±0.03 | 0.11±0.04 | 0.12±0.03 | 0.12±0.06 | 0.12±0.03 |
| 13:0 | 0.45±0.15[b] | 0.37±0.14[ab] | 0.25±0.16[a] | 0.43±0.22[ab] | 0.43±0.15[b] |
| 14:0 | 3.58±0.57 | 3.14±0.53 | 3.29±0.65 | 3.18±0.69 | 3.15±0.83 |
| 15:0 | 1.16±0.17 | 1.04±0.27 | 1.09±0.16 | 1.04±0.36 | 1.05±0.25 |
| 16:0 | 16.07±2.31 | 15.51±3.28 | 15.73±3.63 | 15.98±4.57 | 16.03±4.14 |
| 17:0 | 1.48±0.53[b] | 1.36±0.44[ab] | 1.08±0.37[a] | 1.46±0.45[ab] | 1.47±0.29[ab] |
| 18:0 | 9.58±1.39[b] | 6.07±2.33[a] | 8.31±2.68[b] | 5.91±2.14[a] | 5.02±1.29[a] |
| 20:0 | 1.06±0.32 | 1.02±0.45 | 1.04±0.31 | 1.06±0.53 | 1.06±0.26 |
| 21:0 | 0.58±0.19 | 0.55±0.21 | 0.57±0.31 | 0.57±0.31 | 0.58±0.17 |
| 22:0 | 1.07±0.19 | 1.01±0.26 | 1.05±0.42 | 1.05±0.59 | 1.04±0.33 |
| 23:0 | 0.63±0.17 | 0.59±0.25 | 0.62±0.61 | 0.58±0.34 | 0.60±0.22 |
| 24:0 | 1.13±0.35 | 1.06±0.53 | 1.11±0.34 | 1.09±0.49 | 1.08±0.38 |
| 14:1n5 | 0.68±0.24[ab] | 0.62±0.25[ab] | 0.41±0.27[a] | 0.72±0.40[ab] | 0.73±0.26[b] |
| 16:1n7 | 2.93±1.07[b] | 2.54±1.38[ab] | 2.62±1.06[ab] | 1.97±0.55[a] | 2.13±0.40[ab] |
| 18:1n9t | 1.00±0.29 | 0.92±0.31 | 0.67±0.35 | 1.06±0.51 | 1.06±0.35 |
| 18:1n9c | 15.28±2.92 | 16.24±3.82 | 12.91±6.75 | 12.97±4.75 | 15.08±4.19 |
| 20:1 | 4.93±1.37[a] | 5.98±2.25[ab] | 7.41±2.66[b] | 5.61±1.71[ab] | 5.58±1.45[ab] |
| 22:1n9 | 0.90±0.16[a] | 1.81±1.34[ab] | 3.47±4.70[b] | 1.86±1.22[ab] | 1.65±0.34[ab] |
| 24:1n9 | 1.56±0.40[a] | 1.77±0.30[ab] | 1.98±0.78[ab] | 2.11±0.57[b] | 2.02±0.33[ab] |
| 18:2n6t | 1.16±0.44[ab] | 1.42±0.74[ab] | 0.76±0.64[a] | 1.66±1.12[b] | 1.74±0.72[b] |
| 18:2n6c | 1.27±0.14 | 1.46±0.35 | 1.31±0.35 | 1.43±0.5 | 1.57±0.61 |
| 18:3n6 | 0.59±0.20[ab] | 0.89±0.50[ab] | 0.46±0.43[a] | 1.06±0.74[b] | 1.12±0.47[b] |
| 18:3n3 | 0.85±0.17 | 1.22±0.34 | 0.98±0.34 | 1.29±0.68 | 1.33±0.44 |
| 20:2 | 1.09±0.16[a] | 1.28±0.28[ab] | 1.24±0.28[ab] | 1.64±0.44[c] | 1.54±0.36[bc] |
| 20:3n6 | 0.50±0.13[a] | 0.86±0.43[ab] | 0.52±0.36[a] | 1.03±0.64[b] | 1.08±0.42[b] |
| 20:4n6 | 1.79±0.50[a] | 2.17±1.49[ab] | 1.71±1.39[a] | 2.82±1.81[ab] | 3.08±1.03[b] |
| 22:2n6 | 0.62±0.21[ab] | 1.03±0.61[abc] | 0.52±0.51[a] | 1.25±0.87[bc] | 1.31±0.55[c] |
| 20:5n3 | 6.32±0.93[ab] | 5.55±1.35[a] | 7.04±1.55[b] | 6.24±1.82[ab] | 5.26±0.92[a] |
| 22:6n3 | 20.95±3.90 | 21.33±2.19 | 20.64±3.92 | 21.1±4.68 | 20.58±3.06 |
| *Main FA Classes (%ΣTFA)* | | | | | |
| ΣSFA | 37.24±2.81[b] | 32.05±4.20[a] | 34.44±3.50[ab] | 32.73±4.16[a] | 31.86±4.48[a] |
| ΣMUFA | 27.29±4.13 | 30.61±5.01 | 29.76±4.28 | 27.53±4.20 | 29.33±4.09 |
| ΣPUFA | 35.47±5.03 | 37.33±4.57 | 35.80±4.50 | 39.74±4.85 | 38.81±5.35 |
| *Total fatty acids (mg/g dry weight)* | | | | | |
| ΣTFA | 141.56±21.8 | 150.48±18.31 | 147.08±13.66 | 145.68±17.44 | 154.5±15.10 |

±0.22°C) and the lowest in November (13.44±0.74°C) (Fig 4A). Monthly sea surface chlorophyll-*a* (Chl-*a*) also varied significantly among months ($F = 10.92$, $P<0.01$), being the lowest in August (mean±SD, 0.25±0.02 mg m$^{-3}$) and the highest in October (mean±SD, 0.61±0.07 mg m$^{-3}$) (Fig 4B).

There were no significant effects of sea surface temperature on either $\delta^{15}$N or $\delta^{13}$C (GAMM, $\delta^{15}$N, $F = 0.00$, $P = 0.92$; $\delta^{13}$C, $F = 0.00$, $P = 0.67$), nor were there significant effects of

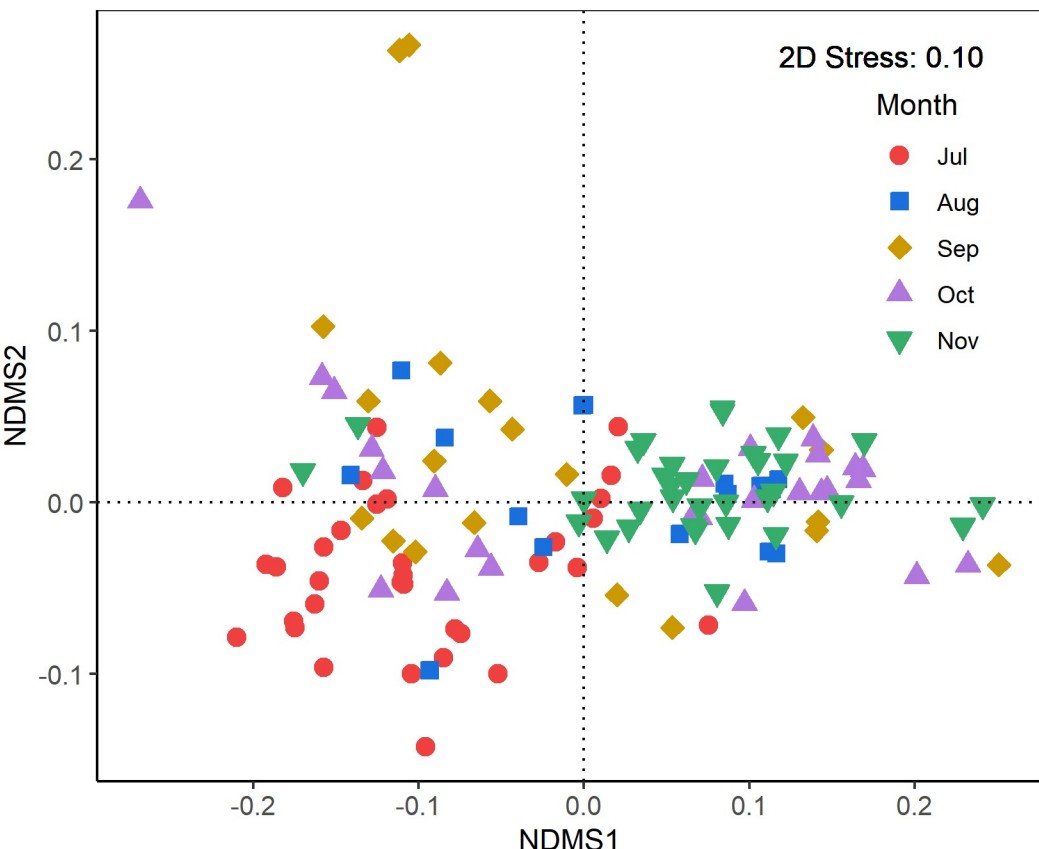

**Fig 3. Non-metric multidimensional scaling (nMDS) ordination based on the Bray–Curtis dissimilarity metric showing overlap in the fatty acid composition from different sampling months.**

chlorophyll-*a* on these isotopic ratios (GAMM, $\delta^{15}$N, $F = 0.06$, $P = 0.22$; $\delta^{13}$C, $F = 0.00$, $P = 0.55$) (S1 Table). For those individual fatty acids that differed significantly among months (Table 2), only 24:1n9 and 20:4n6 were significantly related to the Chl-*a* (S2 Table), with their amounts increasing with increasing Chl-*a* (Fig 5).

## Discussion

Our work indicates that neon flying squid, *O. bartramii*, can provide information about the prey community in the southwestern part of the Western Subarctic Gyre in the northwest Pacific Ocean. *Ommatrephes bartramii* is an appropriate biological sampler for this region because it feeds throughout the water column [15], the digestive gland provides information

**Table 4. Results of analysis of similarities (ANOSIM) for the change in fatty acid compositions between months.** The ANOSIM R value ranges from -1 to 1, where a 1 indicates complete difference between groups, and 0 indicates high similarity.

| Terms | *R*-value | *P*-value |
|---|---|---|
| July *vs.* August | 0.18 | 0.024 |
| August *vs.* September | 0.08 | 0.057 |
| September *vs.* October | 0.10 | 0.040 |
| October *vs.* November | 0.10 | 0.003 |
| pooled | 0.15 | 0.001 |

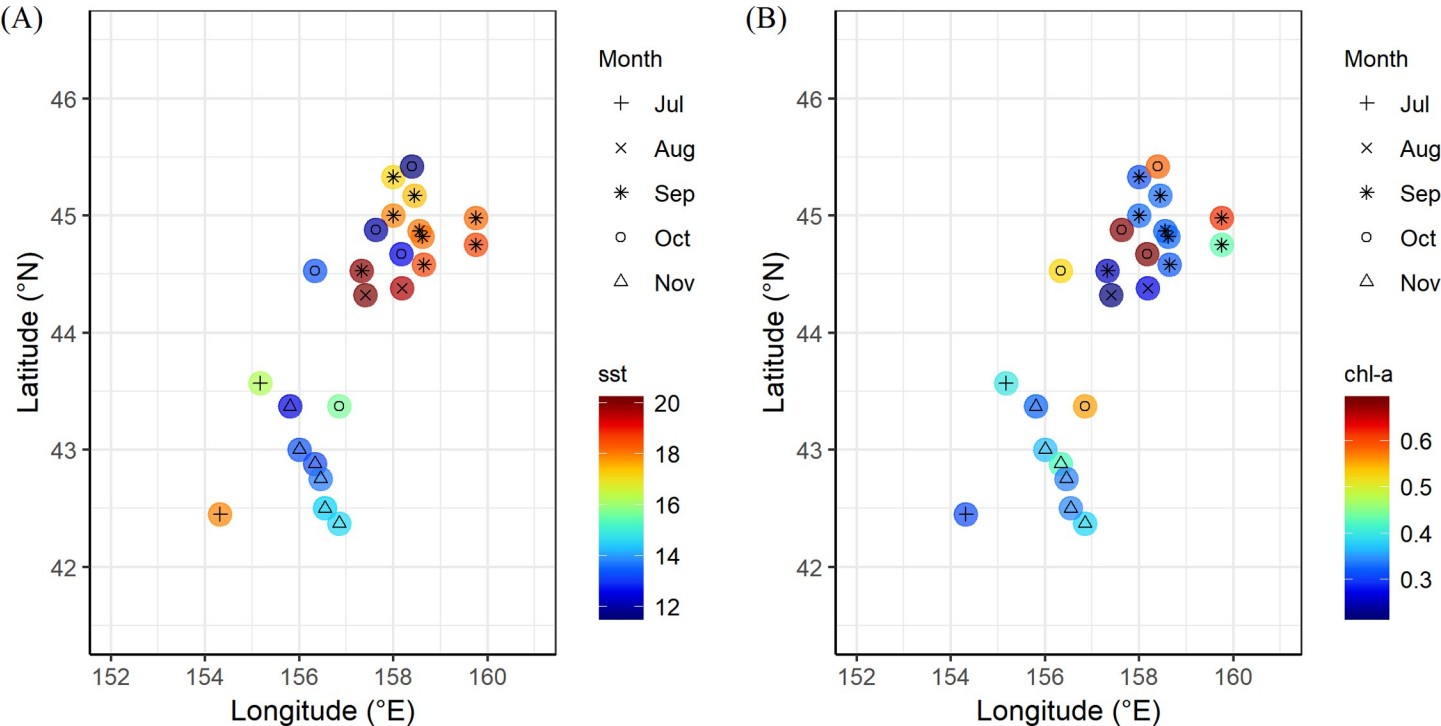

**Fig 4. Monthly mean sea surface temperature at the sampling stations in western subarctic gyre of northwest pacific ocean from July to November 2016.** (A) Sea surface temperature (SST) and (B) Sea surface chlorophyll-*a* concentration (Chl-*a*).

about recent feeding [45–48], and lipids are stored with little or no modification [45, 68, 69]. To our knowledge, this is the first study that uses squid as a biological sampler, combined with the use of stable isotopes and fatty acids to explore trophic dynamics for an oceanic ecosystem.

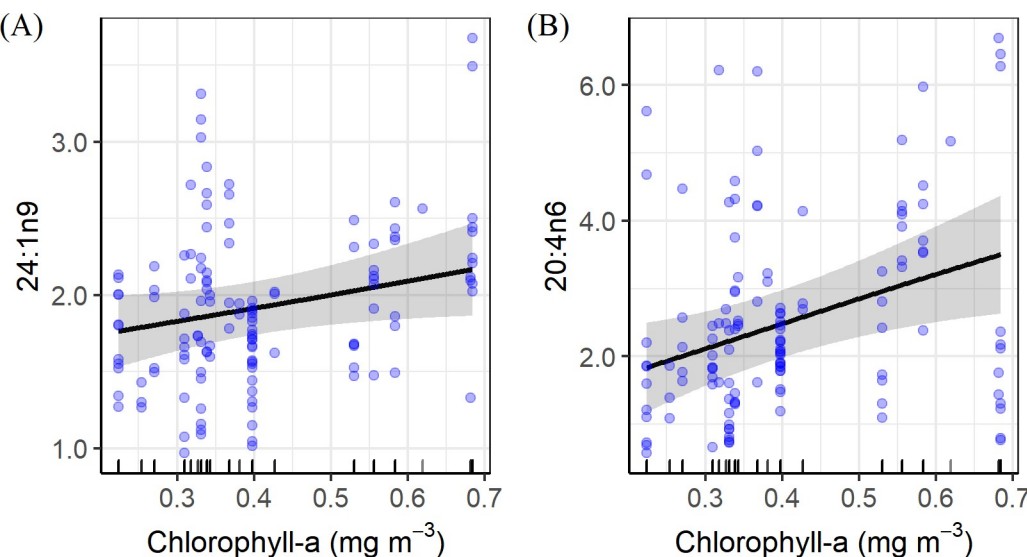

**Fig 5.** Smooth plots from generalized additive mixed models (GAMMs) showing the significant influence of sea surface chlorophyll-*a* concentration (Chl-*a*) on fatty acids 24:1n9 (A) and 20:4n6 (B). Solid line is the estimate of the smooth, grey shade represents 95% confidence intervals, and blue circles represent the raw data.

The similar pattern of isotopic values and the obvious overlap of the fatty acids reveal a stable trophic structure of the prey species community during the second half of the year in this region. Specifically, the $\delta^{13}C$ and $\delta^{15}N$ ratios did not change significantly during this period. The $\delta^{13}C$ is effective for determining foraging habitat [34, 36], and the non-significant differences could reflect that the prey species share similar habitats in the region, regardless of temporal fluctuations. On the other hand, the similar pattern of $\delta^{15}N$ could indicate that a stable prey community was available to *O. bartramii*, given that a stable nitrogen isotope ratio can be effective at identifying the trophic structure for marine organisms [34, 36]. The Western Subarctic Gyre is characterized by highly diverse species and abundant biomass, particularly in the margin areas of the gyre [36, 52, 54], and it is one of the most important feeding grounds for many higher trophic level species, including *O. bartramii* [19, 42, 70]. A stable prey community appears to be vital to support the large populations of these species in this region [11, 42, 71].

Prey availability, to a lesser extent, is responsible for the trophic characteristics of organisms [2]. As such, the stability of $\delta^{15}N$ over months, along with similar niche widths would be expected if the trophic dynamics of the prey community were stable throughout the five sampling months. The variance in the isotope space is an integrated measure of niche width and reflects the variation in the diets of consumers [34, 61, 72]. The dynamics of prey species will tend to result in the highest variance of isotopic niche space in a given ecosystem [73], and ultimately determine the isotopic niche width of the consumers [34, 61]. It is documented that the stomach contents of *O. bartramii* effectively reflect the prey availability locally, and for example, indicate the endemic species *Maurolicus imperatorius* in the transitional zone of the Central North Pacific in July [18] and the migratory myctophids such as *Engraulis japonicas* and *Watasenia scintillans* in the Kuroshio–Oyashio transition of the western North Pacific during the Autumn and early Spring [71]. The $\delta^{15}N$ values in the mantle muscle of *O. bartramii* from the northern part of the Central North Pacific showed moderate variation, due to the prey items prevalently composed by myctophid *Symbolophorus evermanni* and squid families Onychoteuthidae and Enoploteuthidae [17, 41, 43]. Therefore, it would be not unexpected that the prey community occupies a similar niche space and does not temporally change over the sampling months in the southwestern part of the gyre.

The obvious overlap of fatty acid compositions further supports the inference of the stability of the prey community. The multivariate analyses showed that the fatty acid compositions in different sampling months are very similar, evidenced by the clear overlap of the nMDS scatterplots (Fig 3) and low ANOSIM statistic R values for each two consecutive sampling month period (Table 4). These observations suggest that the prey community is composed of either single species or many species that consistently occur in the gyre region throughout July to November, as the fatty acids in higher-order consumers match their diets [47, 48, 74, 75]. There is no reasonable evidence that *O. bartramii* would prey on a single species, because the variation of $\delta^{15}N$ in the digestive gland is larger than the typical enrichment of the nitrogen isotope per trophic level (about 3‰ per trophic level [32]). Indeed, *O. bartarmii* is a well-known voracious generalist that preys on many food items (e.g., Watanabe et al.[18]), and exhibits more variation of nitrogen isotopes than a typical trophic level [19, 76].

Nearly half of the individual fatty acids varied significantly between months (Table 3). In marine environments, many fatty acids have been identified as good tracers of distinct taxa. For example, 16:1n7 and 20:5n3 are indicators of first-order carnivores, 16:0, 18:0 and 22:6n3 of second-order carnivores, and 20:4n6 and 22:4n6 of top predators [38]. 18:0 is also an important tracer of herbivores, and 22:4n6 of planktivores [38]. Accordingly, the lowest values of 16:1n7 in October and 20:5n3 in November may imply that the first-order carnivores were at lower abundance during these months. By contrast, the top predators in the prey community could be much more abundant in November, as suggested by the high value for 20:4n6.

Regarding the second-order carnivores, they should be relatively stable from July to November because no significant differences among months were found for 16:0 and 22:6n3. Coupling with the obvious overlap and similarity of the fatty acid compositions (Fig 3; Table 4), such findings highlight that the prey community in the Western Subarctic Gyre is likely to be in dynamic equilibrium. This is because variation among species is essential for ecosystem stability [77]. Large populations of predators including suspension feeders to carnivores seasonally migrate to the subarctic northwest Pacific [13], so a dynamic equilibrium of the prey community would be expected in the southwestern part of the gyre region, and this stability may be maintained by the high productivity of prey species at the same trophic level, along with seasonal fluctuations.

Monthly sea surface temperature (SST) and chlorophyll-*a* (Chl-*a*) differed significantly among months at the sampling stations. However, our findings indicate that the general pattern of isotopic values was not correlated with the environmental variables, suggesting that the trophic structure of the prey community is stable regardless of changes of the ambient environment. Similar results were obtained for the fatty acids that varied significantly from July to November. There was no evidence that the variation of the individual fatty acids was a function of SST or Chl-*a*, with the exception of 24:1n9 and 20:4n6 (Fig 5). These findings seem to contradict the general arguments about marine species responding to oceanic environments. For example, populations or species may differ in their life-history traits (e.g. growth rate) and subsequent biomass due to changes in water temperature and/or primary productivity (indication through Chl-*a*) [78]. However, it is noteworthy that life-history traits and subsequent ecology for individual species may depend on community composition and demography [6, 79]. First, conditions become more favorable for some species and less favorable for others [1, 80], thereby influencing a species' ecological relevance and ultimately altering the prey species available for top predators [27, 28]. Second, life-history traits and the diversity expressed within species are evolutionarily flexible, such that shifts in life-history strategies, such as staggered age structure, may reduce the risk that an entire cohort will encounter unfavorable environmental conditions [81]. Such flexibility may enable the community to be more resilient to environmental variation. Finally, the physiological tolerance of a species to ambient environmental conditions might increase with its ontogeny [2, 14, 23]. This ultimately contributes to the dynamic equilibrium of a trophic community. Cumulatively, the stable trophic community in the southwestern part of the gyre may have evolved to have high resilience to the regional environment, possibly through high productivity or shifts in life-history strategies among species.

The zooplankton-based food web in the western subarctic Pacific [54, 55] may be another reason for the stability of the trophic community. Large interzonal copepods predominate the zooplankton assemblage [55], and many mesozooplankton including copepods, euphausiids and salps are spatially patchy, creating local zones of high prey availability for predators [54]. However, more detailed information on population connectivity, interaction webs, and structure-forming species is necessary to specifically examine the stability of the community. Additionally, although our survey covered a relatively long time period from July to November, information about the prey community in other seasons is lacking. Further work is needed to address the status of the prey community in other seasons, even though the life history patterns of some important copepods in the gyre may be independent of variable environments [55].

## Conclusions

We demonstrate that stable isotopes and fatty acid composition data from the digestive gland of *Ommatrephes bartramii*, an opportunistic top predator, varies little from July to November in the southwestern part of the Western Subarctic Gyre of the northwest Pacific Ocean. These

findings imply a stable prey community in the gyre region. The prey community may be resilient to fluctuations in the environment due to high productivity within trophic levels and shifts in life-history strategy with ontogeny. Although trophic analyses at the taxonomic level are still necessary to evaluate the dynamics of prey communities, our work enhances understanding of trophic dynamics in this region, and highlights the use of top predators as biological samplers to better understand trophic dynamics. Voracious and active top predators, combined with stable isotopes and fatty acid techniques can provide trophic information at multiple time scales, allowing an assessment of trophic dynamics. This methodology should be generally applicable to an oceanic system that is poorly sampled.

## Supporting information

**S1 Table. GAMM results for stable isotopes ($\delta^{15}$N and $\delta^{13}$C) for *Ommastrephes bartramii* modeled in relation to monthly mean sea surface temperature (SST) and chlorophyll a concentration (Chl-*a*) in the western subarctic gyre of northwest pacific ocean.** AIC, Akaike information criterion; BIC, Bayesian information criterion; DF, degree of freedom; logLik., maximum log-likelihood ratio; edf, estimated degrees of freedom; Ref.df, reference degree of freedom (prior to deductions); R-sq.(adj), adjusted R-squared; Std.Dev., standard deviation; Std.Error, standard error.
(DOCX)

**S2 Table. GAMM results for fatty acids of *Ommastrephes bartramii* modeled in relation to monthly mean sea surface temperature (SST) and chlorophyll a concentration (Chl-*a*) in the western subarctic gyre of northwest pacific ocean.** The fatty acids used for GAMM are those differed significantly between sampling months (details see Table 1). Akaike information criterion; BIC, Bayesian information criterion; DF, degree of freedom; logLik., maximum log-likelihood ratio; edf, estimated degrees of freedom; Ref.df, reference degree of freedom (prior to deductions); R-sq.(adj), adjusted R-squared; Std.Dev., standard deviation; Std.Error, standard error.
(DOCX)

## Acknowledgments

This is a contribution of the Distant Squid Fisheries Sci-Tech Group (SHOU). We thank the staff members of the Key Laboratory of Sustainable Exploitation of Oceanic Fisheries Resources, Ministry of Education, Shanghai Ocean University for assistance in the laboratory. We are grateful to technicians Shaoqin Wang and Chunxia Gao for the determination of the fatty acids. We thank Kai Zhu, Fei Han, Sipeng Xuan, Yanran Wei, Zimo Chen for the biological data collected, and Dr. Stuart Corney for his insightful comments. We also appreciate Dr. Andre Punt for his cordial help with English edits and insightful comments.

## Author Contributions

**Conceptualization:** Dongming Lin, Xinjun Chen.

**Data curation:** Dongming Lin.

**Formal analysis:** Dongming Lin.

**Funding acquisition:** Dongming Lin, Xinjun Chen.

**Investigation:** Dongming Lin.

**Methodology:** Dongming Lin, Xinjun Chen.

**Project administration:** Xinjun Chen.

**Resources:** Dongming Lin.

**Software:** Dongming Lin.

**Supervision:** Dongming Lin, Xinjun Chen.

**Visualization:** Dongming Lin.

**Writing – original draft:** Dongming Lin, Xinjun Chen.

**Writing – review & editing:** Dongming Lin, Xinjun Chen.

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
