## [Decision Letter · Decision Letter 0]

24 Feb 2020

PONE-D-19-29709

Top predator reveals the stability of prey-community in the western subarctic Pacific

PLOS ONE

Dear Dr. Lin,

Thank you for submitting your manuscript to PLOS ONE. After careful consideration, we feel that it has merit but does not fully meet PLOS ONE’s publication criteria as it currently stands. Therefore, we invite you to submit a revised version of the manuscript that addresses the points raised during the review process.

In your revision address all the comments and suggestions made by reviewer #1, and in particular revise your usage of English. 

We would appreciate receiving your revised manuscript by Apr 09 2020 11:59PM. To enhance the reproducibility of your results, we recommend that if applicable you deposit your laboratory protocols in protocols.io, where a protocol can be assigned its own identifier (DOI) such that it can be cited independently in the future. For instructions see: http://journals.plos.org/plosone/s/submission-guidelines#loc-laboratory-protocols

We look forward to receiving your revised manuscript.

Kind regards,

Andrea Belgrano, Ph.D.

Academic Editor

PLOS ONE

Journal Requirements:

Reviewers' comments:

Reviewer's Responses to Questions

**Comments to the Author**

1. Is the manuscript technically sound, and do the data support the conclusions?

Reviewer #1: Yes

2. Has the statistical analysis been performed appropriately and rigorously? 

Reviewer #1: Yes

3. Have the authors made all data underlying the findings in their manuscript fully available?

Reviewer #1: Yes

4. Is the manuscript presented in an intelligible fashion and written in standard English?

Reviewer #1: No

5. Review Comments to the Author

Reviewer #1: Lin and Chen use isotopic ratios from top predators to evaluate the stability of prey communities in the sub-Arctic ocean.

General Comments

This study is interesting, rooted in strong theory and logical predictions, and in an interesting and difficult to access part of the world. However, the writing style and readability of the paper are deficient, I wonder if this has been critically reviewed by a native English speaker? If not that would be a good first step of a thorough rewrite. As it stands, the MS is difficult to read due to widespread grammatical and spelling errors, often hindering appreciation of the science being reported. I note several instances in my specific comments, but the authors should review the manuscript thoroughly with a native English speaker.

The abstract lacks critical background information: over what time period and at what frequency were biological and environmental measurements made? To what degree did SST and chlorophyll vary over the course of the study, enough to make us expect that diets should be changing as well?

To what extent do the months sampled create bias in the findings of the paper? I would guess that July – November is the warmest and perhaps most stable period of ocean temperatures in the region. Might the prey community and isotopic values change more during winter months?

Specific Comments

Line 18, 21, and elsewhere: prey community (remove -)

Line 20: squid Ommastrephes bartramii, an active top marine predator, as a biological sampler to investigate the…

Line 23 and throughout the manuscript: Entire MS needs thorough revision by a native English speaker, for instance this sentence reads much better as “No significant differences in stable isotope ratios were detected among sampling months.”

Please state the period of time over which sampling occurred

Line 38: stability and subsequent service of ecosystems [3-4]. As stability is the central to an ecosystem

Here and throughout the manuscript are many out of place “the”, mistakes with plurals, etc.

Line 72: Alternative to what? Please give examples and how they compare

Line 74: Exemplary run-on sentence without clear structure, requiring substantial re-write by native English speaker.

Line 93: Apart from what? Do you mean trophic level in width? The cited study found that O. bartramii is substantially more than a trophic level apart from another species of squid, S. oualaniensis, which you do not bring up here.

Line 95 and Line 340-343: This is the one critical concern I have about the manuscript regarding methodology and assumptions, does a change in diet necessarily indicate prey stability? Could the squid change its feeding behavior, location, or timing to keep up with changes in prey distributions? For instance, their daily vertical migrations integrate across several ocean ecosystems, and prey could become scarce at one depth and abundant at another? Are you assuming that predation rate and prey abundance are density-dependent?

Line 105: still exist

Line 113: Owing to closing? This is not clear

Line 182: Here or in results for this method, indicate what a normal non-overlapping SEAc proportion looks like, what do high or low values tell you? Orientate your reader to the method

Line 234: evidenced by the (large?) non-overlap (SEAc) proportion (Table 2) and the high degree of overlap for ellipses of different months

Line 246: No significant difference in total fatty acid values was found among sample months (ANOVA…

Line 381: This is a nice theoretical paragraph. Please include some discussion of 1) the implications of the open ocean environment (homogeneous, large, low and unpredictable encounter rates with other organisms, etc.) for the use of stable isotopes for inference of prey dynamics, would this method be applicable in other systems? And 2) the implications of this ecosystem being based on micro-zooplankton instead of phytoplankton, as mentioned on Line 116, for community stability. Where do these micro-zooplankton come from, and are they inherently more stable than a phytoplankton-based food web?

6. PLOS authors have the option to publish the peer review history of their article (what does this mean?). If published, this will include your full peer review and any attached files.

Reviewer #1: No

---

## [Author Response · Author response to Decision Letter 0]

4 Apr 2020

Dear Editor and Reviewer,

Thank you so much for your constructive comments and suggestions. 

We have closely followed the comments and suggestions provided by the reviewers when revised the manuscript. Major changes made to the work include:

- Adding information about the sampling time, sampling procedure, and the significant variation of SST and chlorophyll in the abstract;

- Clarifying the methodology and assumptions in the text;

- Including some discussion of stable prey community possibly based on zooplankton.

- Asking native English-speaking colleagues to review thoroughly the revised manuscript.

For details, please see the point-by-point response to all comments and suggestions below.

Reviewer #1: Lin and Chen use isotopic ratios from top predators to evaluate the stability of prey communities in the sub-Arctic ocean.

General Comments

This study is interesting, rooted in strong theory and logical predictions, and in an interesting and difficult to access part of the world. However, the writing style and readability of the paper are deficient, I wonder if this has been critically reviewed by a native English speaker? If not that would be a good first step of a thorough rewrite. As it stands, the MS is difficult to read due to widespread grammatical and spelling errors, often hindering appreciation of the science being reported. I note several instances in my specific comments, but the authors should review the manuscript thoroughly with a native English speaker.

Response: Thank you so much for your recognition of our work. We asked two English-speaking colleagues (based in Australia and the US) to review the revised manuscript and they provided extensive suggestions to improve readability, grammar, and spelling. We trust that the paper is now of the standard expected of the journal.

The abstract lacks critical background information: over what time period and at what frequency were biological and environmental measurements made? To what degree did SST and chlorophyll vary over the course of the study, enough to make us expect that diets should be changing as well?

Response: We have added information about the sampling period as well as the SST and Chl-a data. The samples were collected monthly from July to November 2016 while the SST and Chl-a data were downloaded from the NOAA ERDDAP (Version 1.82), at a monthly mean scale for each sampling station. SST varied from 13.44�0.74 ℃ to 19.83�0.22 ℃, and Chl-a varied from 0.25�0.02 mg m-3 to 0.61�0.07 mg m-3. The statistical analysis indicated that both SST and Chl-a differed significantly among sampling months. Details please see our revised MS.

To what extent do the months sampled create bias in the findings of the paper? I would guess that July – November is the warmest and perhaps most stable period of ocean temperatures in the region. Might the prey community and isotopic values change more during winter months?

Response: Based on the SST and Chl-a data from the NOAA ERDDAP, we know that both environmental variables vary significantly during our survey period (July to November). Monthly mean SST varied from 13.44℃ to 19.83℃, and monthly mean Chl-a varied from 0.25 mg m-3 to 0.61 mg m-3. 

Our data only cover summer and autumn, and the findings might be not applicable to other seasons such as winter when productivity is usually low and spring when primary production is the highest in the Oyashio area in the northwest Pacific. However, the life history strategy of some copepods in the region may be independent of the environment (temperature, presence or absence of the spring bloom and the timing difference of primary production) [1]. Based on your suggestions, we have included some discussion about this issue in the final paragraph of the Discussion section. For details please see our revised MS.

Specific Comments

Line 18, 21, and elsewhere: prey community (remove -)

Response: We have remove the hyphen here and in other places in the MS.

Line 20: squid Ommastrephes bartramii, an active top marine predator, as a biological sampler to investigate the…

Response: We have revised the text following your suggestion.

Line 23 and throughout the manuscript: Entire MS needs thorough revision by a native English speaker, for instance this sentence reads much better as “No significant differences in stable isotope ratios were detected among sampling months.”

Response: We have rephrased it, and asked a native English speaker to edit our manuscript. 

Please state the period of time over which sampling occurred

Response: We have revised it as “Squid were collected monthly from July to November 2016. There were no significant differences among months in stable isotopes (δ13C and δ15N) in the digestive gland, a fast turnover organ reflecting recent dietary information.”

Line 38: stability and subsequent service of ecosystems [3-4]. As stability is the central to an ecosystem

Response: We have rephrased it.

Here and throughout the manuscript are many out of place “the”, mistakes with plurals, etc.

Response: These types of errors have been corrected in the revised version of the manuscript.

Line 72: Alternative to what? Please give examples and how they compare

Response: Sorry for this lack of clarity. Based on your suggestion, we have revised the text to “Biochemical tracers are considered to be a complementary or even alternative and cost-effective tool to stomach content analysis for examining major changes in trophic structure and ecosystem productivity.” Also, we have outlined one example that used fatty acid profiles to reveal temporal dietary shifts in Todarodes filippovae related to site-specific oceanography and ecosystem structure in the continental slope waters in the Southern Ocean. For details please see our revised MS.

Line 74: Exemplary run-on sentence without clear structure, requiring substantial re-write by native English speaker.

Response: We have revised this sentence and other sentences that seem to have similar problems.

Line 93: Apart from what? Do you mean trophic level in width? The cited study found that O. bartramii is substantially more than a trophic level apart from another species of squid, S. oualaniensis, which you do not bring up here.

Response: Yes, you are right. However, we have revised this paragraph following your suggestions below, and this text has been removed. For details, please see our revised MS.

Line 95 and Line 340-343: This is the one critical concern I have about the manuscript regarding methodology and assumptions, does a change in diet necessarily indicate prey stability? Could the squid change its feeding behavior, location, or timing to keep up with changes in prey distributions? For instance, their daily vertical migrations integrate across several ocean ecosystems, and prey could become scarce at one depth and abundant at another? Are you assuming that predation rate and prey abundance are density-dependent?

Response: Thanks for your insightful comments and suggestions. We have added the following text to address this key issue: “More importantly, O. bartramii is a high trophic level species, with an average δ15N value up to 13.6‰ [2-5], which occupies a similar trophic position as other top predators such as albatrosses (mean δ15N, 12.0‰ for Diomedea immutabilis; 14.4‰ for Diomedea nigripes) [2], and sharks (Prionace glauca, mean 12.1‰) [6]. O. bartramii is an opportunistic generalist that preys on a wide variety of species, including crustaceans, fishes and cephalopods [3, 7-9]. The diet of O. bartramii varies spatial-temporally given the associated prey community, e.g., it may feed on transitional-water species during its northward feeding migration [8], migratory mesopelagic species in the epipelagic zone at night [10] and non-migratory species during the day in the mesopelagic zone [8]. O. bartramii therefore has the potential to be an ideal trophic indicator of ecosystem functioning [11], and represents a way integrate the ecological dynamics [12].”

We agree. This species shifts diet with ontogeny, possibly driven by changes in body size and spatial-temporal distribution [9]. However, feeding behavior appears to be associated with prey composition. For instance, O. bartramii in the central North Pacific appears to mainly feed on transitional-water species such as Symbolophorus californiensis and Onychoteuthis borealijaponica during northward feeding migrations from the transitional zone to the transitional domain from May to July, since the seasonal migration patterns of prey and predators are similar spatial-temporally [8]. Additionally, although this species has been recognized as an epipelagic feeder and feeds at night when the bulk of the mesopelagic prey species migrate into the epipelagic layers [10], squids of larger size also feed on non-migratory species such as Protomyctophum thompsoni and Bathylagus ochotensis during the day in the mesopelagic zone where they vertically migrate to [8].

Therefore, O. bartramii has the potential to be a trophic indicator of the dynamics of prey communities within an ecosystem. Also, due to their abundance, O. bartramii has a major impact on the structure of the food web via top-down effects on prey [5, 13]. Hence, to a lesser extent, studies on the dynamics of species preyed on by O. bartramii can determine the nature of the ecosystem they encounter.

Based on your suggestions and concerns, we have carefully revised these sentences. Hopefully the modifications and descriptions above address your concerns.

Line 105: still exist

Response: We have cut down the sentence, so this term has been deleted.

Line 113: Owing to closing? This is not clear

Response: Corrected. We have deleted the word “closing”.

Line 182: Here or in results for this method, indicate what a normal non-overlapping SEAc proportion looks like, what do high or low values tell you? Orientate your reader to the method

Response: Sorry. We have miswritten this sentence, and we have revised it to: “the overlap as the proportion of the sum of the non-overlapping ellipse areas (non-overlap SEAc proportion) based on 1,000 replications”. We have also included some text about the meaning of non-overlapping SEAc, which ranges from 0 when there is no overlap in isotopic niche widths among groups to 1 when there is complete overlap in the isotopic niche widths between groups.

Line 234: evidenced by the (large?) non-overlap (SEAc) proportion (Table 2) and the high degree of overlap for ellipses of different months

Response: Correct, it is the large non-overlap (SEAc) proportion. Accordingly, we have followed your suggestion and revised it to “confirmed by the high non-overlap (SEAc) proportions (Table 2) and the considerable overlap of the ellipses for the different months (Fig 2 B)”.

Line 246: No significant difference in total fatty acid values was found among sample months (ANOVA…

Response: Agreed, we have revised it.

Line 381: This is a nice theoretical paragraph. Please include some discussion of 1) the implications of the open ocean environment (homogeneous, large, low and unpredictable encounter rates with other organisms, etc.) for the use of stable isotopes for inference of prey dynamics, would this method be applicable in other systems? And 2) the implications of this ecosystem being based on micro-zooplankton instead of phytoplankton, as mentioned on Line 116, for community stability. Where do these micro-zooplankton come from, and are they inherently more stable than a phytoplankton-based food web?

Response: Based on your suggestions, we have included some discussion about the zooplankton-based food web in the last paragraph of the Discussion in the revised MS. In the Gyre, the zooplankton assemblage is dominated by large interzonal copepods [1]. The zooplankton feeds and grows at the surface layer for a few months, and resides at depth for remaining months, with 1 or 2 years life cycles [1]. Meanwhile, many mesozooplankton including copepods, euphausiids and salps show intensive patchiness in their spatial distributions, which create local zones of high prey availability for predators [14]. However, it is not possible to know whether a zooplankton-based food web is more stable than a phytoplankton-based food web here. More work is needed to address the hypothesis. Also, we have divided this pargragph into three as it is too large.

Yes, you are right, and the methodology of stable isotopes can be used to infer prey dynamics in other systems. This is because stable nitrogen isotope ratios (δ15N) can be used to identify trophic positions with enrichment of about 3‰ per trophic level [15, 16]. On the other hand, the stable carbon isotope ratio (δ13C) is effective for determining dietary sources, because the ratios in food sources and consumers in higher trophic levels are similar [15, 16]. Regarding fatty acids, many of them can only be biosynthesized by certain phytoplankton and macroalgae species and become essential dietary components to higher trophic levels, where the fatty acids are assimilated without, or with only minimal, modification [17]. Therefore, it is possible to investigate the prey dynamics of a predator by combining these methods.

We have not included text about using stable isotopes to infer prey dynamics for this open ocean system. This is because this paragraph is primarily to illustrate possible reasons why the prey community is stable under a variable environment. Hopefully you agree with us. 

Response: We have uploaded the figure files to PACE, which indicates that all the figures meet PLOS requirements.

Reference

1. Tsuda A, Saito H, Kasai H. Life histories of Eucalanus bungii and Neocalanus cristatus (Copepoda: Calanoida) in the western subarctic Pacific Ocean. Fish Oceanogr. 2004;13(s1):10-20. doi: 10.1111/j.1365-2419.2004.00315.x.

2. Gould P, Ostrom P, Walker W. Trophic relationships of albatrosses associated with squid and large-mesh drift-net fisheries in the North Pacific Ocean. Canadian Journal of Zoology. 1997;75(4):549-62. doi: 10.1139/z97-068.

3. Parry MP. The trophic ecology of two ommastrephid squid species, Ommastrephes bartramii and Sthenoteuthis oualaniensis, in the north Pacific sub-tropical gyre. Manoa: University of Hawaii; 2003.

4. Parry M. Trophic variation with length in two ommastrephid squids, Ommastrephes bartramii and Sthenoteuthis oualaniensis. Mar Biol. 2008;153(3):249-56. doi: 10.1007/s00227-007-0800-3.

5. Kato Y, Sakai M, Nishikawa H, Igarashi H, Ishikawa Y, Vijai D, et al. Stable isotope analysis of the gladius to investigate migration and trophic patterns of the neon flying squid (Ommastrephes bartramii). Fish Res. 2016;173:169-74. https://doi.org/10.1016/j.fishres.2015.09.016.

6. Fujinami Y, Nakatsuka S, Ohshimo S. Feeding Habits of the Blue Shark (Prionace glauca) in the Northwestern Pacific Based on Stomach Contents and Stable Isotope Ratios. Pacific Science. 2018;72(1):21-39. doi: 10.2984/72.1.2.

7. Matthew P. Feeding behavior of two ommastrephid squids Ommastrephes bartramii and Sthenoteuthis oualaniensis off Hawaii. Mar Ecol Prog Ser. 2006;318:229-35. doi: 10.3354/meps318229.

8. Watanabe H, Kubodera T, Ichii T, Kawahara S. Feeding habits of neon flying squid Ommastrephes bartramii in the transitional region of the central North Pacific. Mar Ecol Prog Ser. 2004;266:173-84. doi: 10.3354/meps266173.

9. Jereb P, Roper CFE. Cephalopods of the world. An annotated and illustrated catalogue of cephalopod species known to date. Volume 2. Myopsid and Oegopsid Squids. Rome: FAO; 2010. 605 p.

10. Lipiński MR, Linkowski TB. Food of the squid Ommastrephes bartramii (Lesueur, 1821) from the South-West Atlantic Ocean. S Afr J Mar Sci. 1988;6(1):43-6. doi: 10.2989/025776188784480573.

11. Boyle P, Rodhouse P. Cephalopods: ecology and fisheries. Oxford, UK: Wiley-Blackwell; 2005. 464 p.

12. Bell G, Fortier-Dubois É. Trophic dynamics of a simple model ecosystem. Proceedings of the Royal Society B: Biological Sciences. 2017;284(1862):20171463. doi: 10.1098/rspb.2017.1463.

13. Ichii T, Mahapatra K, Sakai M, Wakabayashi T, Okamura H, Igarashi H, et al. Changes in abundance of the neon flying squid Ommastrephes bartramii in relation to climate change in the central North Pacific Ocean. Mar Ecol Prog Ser. 2011;441:151-64.

14. Mackas DL, Tsuda A. Mesozooplankton in the eastern and western subarctic Pacific: community structure, seasonal life histories, and interannual variability. Progress in Oceanography. 1999;43(2):335-63. https://doi.org/10.1016/S0079-6611(99)00012-9.

15. Post DM. Using stable isotopes to estimate trophic position: models, methods, and assumptions. Ecology. 2002;83(3):703-18. doi: 10.1890/0012-9658(2002)083[0703:USITET]2.0.CO;2.

16. Layman CA, Araujo MS, Boucek R, Hammerschlag-Peyer CM, Harrison E, Jud ZR, et al. Applying stable isotopes to examine food-web structure: an overview of analytical tools. Biol Rev. 2012;87(3):545-62. doi: 10.1111/j.1469-185X.2011.00208.x.

17. Iverson SJ. Tracing aquatic food webs using fatty acids: from qualitative indicators to quantitative determination. In: Kainz M, Brett MT, Arts MT, editors. Lipids in Aquatic Ecosystems. New York, NY: Springer New York; 2009. p. 281-308.

---

## [Decision Letter · Decision Letter 1]

28 Apr 2020

PONE-D-19-29709R1

Top predator reveals the stability of prey community in the western subarctic Pacific

PLOS ONE

Dear Dr. Lin,

Thank you for submitting your manuscript to PLOS ONE. After careful consideration, we feel that it has merit but does not fully meet PLOS ONE’s publication criteria as it currently stands. Therefore, we invite you to submit a revised version of the manuscript that addresses the points raised during the review process.

In your final revision, please address the minor edits and suggestions made by reviewer #1.

We would appreciate receiving your revised manuscript by Jun 12 2020 11:59PM. To enhance the reproducibility of your results, we recommend that if applicable you deposit your laboratory protocols in protocols.io, where a protocol can be assigned its own identifier (DOI) such that it can be cited independently in the future. For instructions see: http://journals.plos.org/plosone/s/submission-guidelines#loc-laboratory-protocols

We look forward to receiving your revised manuscript.

Kind regards,

Andrea Belgrano, Ph.D.

Academic Editor

PLOS ONE

Reviewers' comments:

Reviewer's Responses to Questions

**Comments to the Author**

1. If the authors have adequately addressed your comments raised in a previous round of review and you feel that this manuscript is now acceptable for publication, you may indicate that here to bypass the “Comments to the Author” section, enter your conflict of interest statement in the “Confidential to Editor” section, and submit your "Accept" recommendation.

Reviewer #1: (No Response)

2. Is the manuscript technically sound, and do the data support the conclusions?

Reviewer #1: Yes

3. Has the statistical analysis been performed appropriately and rigorously? 

Reviewer #1: Yes

4. Have the authors made all data underlying the findings in their manuscript fully available?

Reviewer #1: Yes

5. Is the manuscript presented in an intelligible fashion and written in standard English?

Reviewer #1: Yes

6. Review Comments to the Author

Reviewer #1: Second review for “Top predator reveals the stability of prey community in the western subarctic Pacific”

General Comments

Lin and Chen have submitted a considerably improved version of their manuscript. I find very few faults in the English writing (but see a few in specific comments). The manuscript is interesting and thorough and is near ready for publication. I think there could still be a bit more exploration of the limitations of the data and possible interpretations of the surprisingly stable dietary dynamics despite changes in oceanographic parameters. Some further development considering what the squid are actually eating, what features of their ecology promote this apparent ecological stability, how you might see different trends in different predator species or across a larger coverage of annual trends, etc. See below for some suggestions for expanding the discussion.

Specific Comments

Line 18: The stability of the ecosystems; Also this sentence might be better finished: “but changes in the composition and abundance of prey species are poorly understood, especially in open ocean ecosystems.”

Line 38-41: These two sentences are uninformative, and the use of the word “prey” is misleading (almost every animal is prey to some other animal). Suggest re-writing perhaps as “Concern about ecosystem functioning highlights the need for a better understanding of how the composition and abundance of species in natural communities respond to environmental change. For instance, predatory animals are susceptible to reduction or extirpation of available prey due to environmental processes, which undermines the stability of ecosystems and the services they provide.”

Line 54: Squids grow quickly (or rapidly)

Line 64: such as stable isotopes

Line 66: dietary history over a range of temporal scales (since this is more appropriate to the use of different tissues that vary in their turnover rates)

Line 73: match of those of their prey

Line 78: with little or no modification

Line 84: as a biological sampler..

Line 100: suggest: … and represents a way of integrating ecological dynamics over a large area and across several ecosystems that are difficult to study directly.

Line 100: isotope ratios and fatty acids (no comma)

Line 106: suggest: … quantifying community dynamics in response to environmental change.

Line 125 (and elsewhere) indent on first sentence of each paragraph

Line 127: considered to be one of active…

Line 219: Since tables and figures are often seen separated from the main text, consider defining SEAc in the caption

Line 263: What is the relationship between the temperature and Chl a variation observed in the study months in comparison to the total variation observed throughout the year? If you included a wider range of the possible environmental variation and sampling months would you expect to see a different result?

Line 304: There is some room for discussion I think of what the squid are actually eating, perhaps through comparison with other studies that have gut content analysis paired with stable isotope or fatty acid analysis.

Line 325: If they are feeding on a wide range of diet items, would you not expect there to be considerable variation among squid samples in their diet, whether among individuals within a season or between seasons? You could test for differences in multivariate dispersion in the fatty acid nMDS profiles using betadisper() in the vegan package to examine whether variation in diet among individuals varies over time.

Line 352: Indeed this was a finding I did not expect when reading the intro and methods. More could be made of this surprising finding. I imagine much of the stability over time in diet reflects the large area and multiple ecosystems (depths) across which squid integrate their diets. Would you expect different results if you looked at more vertically restricted species? Or more specialist predators?

Line 357: evolutionarily flexible

Line 367: copepods predominates the

Line 372: long time period (no -)

Line 380: part of the Western…

7. PLOS authors have the option to publish the peer review history of their article (what does this mean?). If published, this will include your full peer review and any attached files.

Reviewer #1: No

---

## [Author Response · Author response to Decision Letter 1]

3 Jun 2020

PONE-D-19-29709R1

Top predator reveals the stability of prey community in the western subarctic Pacific

PLOS ONE

Reviewer #1: Second review for “Top predator reveals the stability of prey community in the western subarctic Pacific”

General Comments

Lin and Chen have submitted a considerably improved version of their manuscript. I find very few faults in the English writing (but see a few in specific comments). The manuscript is interesting and thorough and is near ready for publication. I think there could still be a bit more exploration of the limitations of the data and possible interpretations of the surprisingly stable dietary dynamics despite changes in oceanographic parameters. Some further development considering what the squid are actually eating, what features of their ecology promote this apparent ecological stability, how you might see different trends in different predator species or across a larger coverage of annual trends, etc. See below for some suggestions for expanding the discussion.

Response: Thank you so much for your insightful comments. We have revised the MS based on your constructive suggestions. Details please see our revised version.

Specific Comments

Line 18: The stability of the ecosystems; Also this sentence might be better finished: “but changes in the composition and abundance of prey species are poorly understood, especially in open ocean ecosystems.”

Response: We have revised this sentence following your suggestion.

Line 38-41: These two sentences are uninformative, and the use of the word “prey” is misleading (almost every animal is prey to some other animal). Suggest re-writing perhaps as “Concern about ecosystem functioning highlights the need for a better understanding of how the composition and abundance of species in natural communities respond to environmental change. For instance, predatory animals are susceptible to reduction or extirpation of available prey due to environmental processes, which undermines the stability of ecosystems and the services they provide.”

Response: We have revised the sentences following your suggestions.

Line 54: Squids grow quickly (or rapidly)

Response: We have revised it as “Squids grow rapidly”.

Line 64: such as stable isotopes

Response: We have revised it as “stable isotopes”.

Line 66: dietary history over a range of temporal scales (since this is more appropriate to the use of different tissues that vary in their turnover rates)

Response: We have revised it as “a range of temporal scales”.

Line 73: match of those of their prey

Response: We have revised it as “match of those of their prey”.

Line 78: with little or no modification

Response: We have revised it as “with little or no modification”.

Line 84: as a biological sampler..

Response: We have revised it as “as a biological sampler”.

Line 100: suggest: … and represents a way of integrating ecological dynamics over a large area and across several ecosystems that are difficult to study directly.

Response: We have revised it as “… and represents a way of integrating ecological dynamics over a large area and across several ecosystems that are difficult to study directly”. Thanks!

Line 100: isotope ratios and fatty acids (no comma)

Response: We have deleted the comma.

Line 106: suggest: … quantifying community dynamics in response to environmental change.

Response: We have revised it as “… quantifying community dynamics in response to environmental change”.

Line 125 (and elsewhere) indent on first sentence of each paragraph

Response: We have indented the first sentence of each paragraph throughout the MS.

Line 127: considered to be one of active…

Response: We have revised it.

Line 219: Since tables and figures are often seen separated from the main text, consider defining SEAc in the caption

Response: We have defined SEAc and non-overlap SEAc proportion in the table caption.

Line 263: What is the relationship between the temperature and Chl a variation observed in the study months in comparison to the total variation observed throughout the year? If you included a wider range of the possible environmental variation and sampling months would you expect to see a different result?

Response: Accordingly, we have analyzed the relationship between SST and Chl-a among the study months, and the results showed that there is a light correlation between SST and Chl-a (Radi.2=0.32), though it is significant (F=-3.45, P=0.0023). We have also checked the collinearity between these two predictors, and the results indicated that the collinearity is relatively low (VIF=1.54). So, it is suitable to use these two factors simultaneously to predict the potential effects of the ambient environment on the dynamics of the prey community here.

Regarding SST and Chl-a throughout the sampling year, we have analyzed their variation by month within a rectangle region (153°-160°E, 42°-46°N) including all of the sampling stations in the gyre (Figure 1 below). We can find that both SST and Chl-a of the sampling months have a similar change pattern compared to that of the same months of the sampling year. We can also find that although sea surface temperature was the warmest during the sampling months in the region, the primary production (indication by Cha-a) exhibits a very similar variation to that during the first half of the year. It seems like that the primary production in this region is independent of sea surface temperature, possibly due to the nutrient rich [1-3] and the zooplankton-based food web [4, 5] as discussed in the Discussion section. 

Because we have no samples collected from other months of the year (according to the fishery company, there are no fisheries in this region in first half of the year as the harsh condition), we are unable to assure to expect a different result. However, due to the similar change of primary production (indication through Chl-a) for the first and second half of the year, it would be possible to expect that the trophic community could be not too much different from our findings here when the analysis was performed for the whole year. 

Due to lack of sampling data of the first half year, we did not revise much more in this paragraphs. But as your concerns, we have added the result of collinearity in the last paragraph of “Statistical analysis”. Hope you agree with us in this respect.

Figure 1 The distribution of mean SST and mean Chl-a by month of the sampling year. Data are presented as mean�SD.

Line 304: There is some room for discussion I think of what the squid are actually eating, perhaps through comparison with other studies that have gut content analysis paired with stable isotope or fatty acid analysis.

Response: Based on your suggestion, we have revised this paragraph by adding some discussions to justify our finding. We are sorry for that we can’t find any references about the feeding habits of O. bartramii in the Southwestern Subarctic Gyre. However, there are some of such studies from other regions such as the Central North Pacific and the Kuroshio-Oyashio transition of the western North Pacific. So, we cited them to discuss our findings here. 

Based on the previous studies using stomach contents and stable isotopes (unfortunately, no studies using fatty acids yet), we know that the stomach contents of Ommastrephes bartramii can effectively reflect the prey items within a given region, and the δ15N values in the mantle muscle can indicate the variations of the prey community. So, the stability of δ15N and the similarity of niche widths over sampling months would expect that the prey community occupies a similar niche space and does not change in the southwestern part of the gyre. Details please see our revised MS.

Line 325: If they are feeding on a wide range of diet items, would you not expect there to be considerable variation among squid samples in their diet, whether among individuals within a season or between seasons? You could test for differences in multivariate dispersion in the fatty acid nMDS profiles using betadisper() in the vegan package to examine whether variation in diet among individuals varies over time.

Response: Yes, you are right. Indeed, we have found that nearly half of the individual fatty acids varied significantly between sampling months. Based on your suggestions, we have used betadisper() to test the dispersion of fatty acids among the sampling months, and found that the variance of the fatty acids was significant between sampling months (F=6.38, P=0.0001048). Considering the obvious overlap (nMDS scaterplots) and high similarity (ANOSIM R values) of the fatty acids among the sampling months, it is rational to conclude that the prey community in the Western Subarctic Gyre is a dynamic equilibrium due to the importance of variations among species for a given ecosystem stability [6]. Therefore, we have added content of “Coupling with the obvious overlap and similarity of the fatty acid compositions (Fig. 3; Table 4),” before the sentence of “Such findings highlight that the prey community in the Western Subarctic Gyre is likely to be in dynamic equilibrium.” Hope you agree with us in this respect.

Line 352: Indeed this was a finding I did not expect when reading the intro and methods. More could be made of this surprising finding. I imagine much of the stability over time in diet reflects the large area and multiple ecosystems (depths) across which squid integrate their diets. Would you expect different results if you looked at more vertically restricted species? Or more specialist predators?

Response: Sorry for such confusion. In fact, this paragraph is aimed to explain why our findings contradict the general arguments that populations or species may differ in their life-history traits and subsequent biomass as the changes of water temperature and primary productivity. As mentioned in the “response to reviewers” for the first revision (R1), we have divided the original paragraph including this paragraph into three small paragraphs as it is too large. As your concern, we have merged this paragraph with the previous paragraph at this version.

Yes, you are right. The stability of the prey community reflects the large area and multiple ecosystems across which the squid forages. As you mentioned, it would be expected different results if the biological sampler is a specialist predator or lives in a restricted depth range. However, as indicated in the Introduction, Ommastrephes bartramii is characterized by the high trophic level in the food web, opportunistic generalist, spatial-temporally shift diets associated with the prey species community, etc. Thus, based on the results found in this work, we are confident that the trophic community is stable within the ecosystem in the southwestern part of the Western Subarctic Gyre in the northwestern Pacific Ocean.

Line 357: evolutionarily flexible

Response: We have revised it.

Line 367: copepods predominates the

Response: We have revised it.

Line 372: long time period (no -)

Response: We have revised it.

Line 380: part of the Western…

Response: We have revised it.

Response: We have uploaded the figure files to PACE, which indicates that all the figures meet PLOS requirements.

References

1. Yuan X, Talley LD. The subarctic frontal zone in the North Pacific: Characteristics of frontal structure from climatological data and synoptic surveys. Journal of Geophysical Research: Oceans. 1996;101(C7):16491-508. doi: 10.1029/96jc01249.

2. Qiu B. Kuroshio and Oyashio Currents. In: Steele JH, editor. Encyclopedia of Ocean Sciences. Oxford: Academic Press; 2001. p. 1413-25.

3. Imai K, Nojiri Y, Tsurushima N, Saino T. Time series of seasonal variation of primary productivity at station KNOT (44°N, 155°E) in the sub-arctic western North Pacific. Deep-Sea Research II. 2002;49(24):5395-408. https://doi.org/10.1016/S0967-0645(02)00198-4.

4. Mackas DL, Tsuda A. Mesozooplankton in the eastern and western subarctic Pacific: community structure, seasonal life histories, and interannual variability. Progress in Oceanography. 1999;43(2):335-63. https://doi.org/10.1016/S0079-6611(99)00012-9.

5. Tsuda A, Saito H, Kasai H. Life histories of Eucalanus bungii and Neocalanus cristatus (Copepoda: Calanoida) in the western subarctic Pacific Ocean. Fish Oceanogr. 2004;13(s1):10-20. doi: 10.1111/j.1365-2419.2004.00315.x.

6. McCann KS. The diversity–stability debate. Nature. 2000;405(6783):228-33. doi: 10.1038/35012234.

---

## [Editor Report · Decision Letter 2]

5 Jun 2020

Top predator reveals the stability of prey community in the western subarctic Pacific

PONE-D-19-29709R2

Dear Dr. Lin,

We’re pleased to inform you that your manuscript has been judged scientifically suitable for publication and will be formally accepted for publication once it meets all outstanding technical requirements.

Kind regards,

Andrea Belgrano, Ph.D.

Academic Editor

PLOS ONE

Additional Editor Comments (optional):

Thank you for addressing in your revised manuscript all the remaining comments and suggestions made by reviewer #1.

---

## [Editor Report · Acceptance letter]

10 Jun 2020

PONE-D-19-29709R2 

Top predator reveals the stability of prey community in the western subarctic Pacific 

Dear Dr. Lin:

I'm pleased to inform you that your manuscript has been deemed suitable for publication in PLOS ONE. Congratulations! Your manuscript is now with our production department. 

Kind regards, 

on behalf of

Dr. Andrea Belgrano 

Academic Editor

PLOS ONE